# Biomass, Carbon and Nitrogen Partitioning and Water Use Efficiency Differences of Five Types of Alpine Grasslands in the Northern Tibetan Plateau

**DOI:** 10.3390/ijerph192013026

**Published:** 2022-10-11

**Authors:** Liping Cheng, Beibei Zhang, Hui Zhang, Jiajia Li

**Affiliations:** 1College of Tourism and Management, Pingdingshan University, Pingdingshan 467000, China; 2Shaanxi Key Laboratory of Disaster Monitoring and Mechanism Simulating, Baoji University of Arts and Sciences, Baoji 721013, China

**Keywords:** alpine grassland, biomass, C stocks, N stocks, water use efficiency

## Abstract

(1) Background: Grassland covers most areas of the northern Tibetan Plateau along with important global terrestrial carbon (C) and nitrogen (N) pools, so there is a need to better understand the different alpine grassland growth associated with ecosystem C, N storage and water use efficiency (WUE). (2) Methods: The plant biomass and C, N concentrations, stocks and vegetation WUE of five kinds of alpine grassland types were investigated in northern Tibetan Plateau. (3) Results: The results showed that there were significant differences among five types of alpine grasslands in aboveground biomass (AGB), belowground biomass (BGB), total biomass (TB) and root:shoot (R/S) ratio while the highest value of different indices was shown in alpine meadow type (AM). The AGB and BGB partitioning results significantly satisfied the allometric biomass partitioning theory. The C, N concentrations and C/N of the vegetation in AGB and BGB showed significant grassland type differences. The highest C, N stocks of BGB were in AM which was almost six or seven times more than the C, N stocks of AGB in alpine desert type (AD). There were significant differences in *δ^13^C* and intrinsic water use efficiency (WUEi) under five alpine grassland types while the highest mean values of foliar *δ^13^C* and WUEi were in AD. Significant negative correlations were found between WUEi and C, N concentrations, C/N of AGB and soil water content (SWC) while the correlation with BGB C/N was not significant. For AGB, BGB, TB and R/S, there were positive correlations with C, N concentrations of AGB, BGB and SWC while it had significant negative correlations with C/N of BGB. (4) Conclusions: With regard to its types, it is suggested that the AM or AS may be an actively growing grassland type in the northern Tibetan Plateau.

## 1. Introduction

Grasslands are an important vegetation type of about 50 million km^2^ covering about 40% of the earth’s terrestrial area, whose biomass and productivity of vegetation are very important [1]. The environment in Tibetan Plateau is extremely harsh and also very sensitive to climate changes in China [2]. In recent years, it is facing much environmental degradations caused by some intensifying human activities such as overgrazing, Chinese herbal medicine collection and so on [3].

With the grassland biomass and productions being the important factors to determine C, N stocks and bioenergy generation, any biomass and growth changes might lead to some reallocation of biomass which includes both aboveground biomass (AGB) and belowground biomass (BGB), consequently resulting in implications for C and N cycles [4]. AGB and BGB, approximately 10% of global carbon (C) stocks [5] are the major contributors to organic C. Thus, it might significantly affect greenhouse gas emissions and influence the global C cycle [6]. In addition to that, as the renewable energy generation sources, it is stipulated to generate about 56% bioenergy by 2020 [7,8]. Furthermore, N stocks are also linked to the ecosystem C dynamic [9] and their growth during the allocation, availability and turnover processes [10].

The water absorption and assimilation of plants, which are closely related to the soil water content (SWC), are the critical physiological processes, especially in alpine grasslands. Moreover, the changing availability of soil water can influence C, N cycling along the soil–plant–atmosphere continuum (SPAC) system [11]. Water use efficiency (WUE), as an important index of the trade-off between water transpiration and carbon production, is also physiologically linked to carbon isotope discrimination (CID, *Δ*) of plants [12]. Over the past years, some relative research about WUE were studied in the Tibetan Plateau. Yang et al. [13] used wood *δ^13^C* to analyze the influences of climatic changes on WUE of alpine meadows in Tibetan Plateau. The foliar *δ^13^C* had no significant difference with the intrinsic water use efficiency (WUEi) under non-grazing and open-grazing sites which was studied by Wu et al. [14]. Luo et al. [15] researched that there was a close relationship between the alpine grassland NPP and foliar *δ^13^C* in the Qinghai-Tibetan Plateau. The research of Chen et al. revealed the spatial distribution of leaf *δ^13^C* in the northern Tibetan Plateau which increased with increasing altitude and decreasing latitude and longitude [16]. The main focus of previous studies was on the degradations, grazing or climate change effects to plant growth, C, N cycles or WUE under alpine grassland ecosystems in the Tibetan Plateau. However, there is still a lack of knowledge about whether the differences in the grassland types are related to the biomass C, N storage and vegetation WUE differences in the northern Tibetan Plateau. Our study specifically addressed the differences of biomass, allocation and storage of C and N and vegetation WUE between five alpine grassland types and identified the correlations between the WUE and biomass C, N stocks in the northern Tibetan Plateau.

## 2. Materials and Methods

### 2.1. Study Sites and Conditions

The alpine steppe (AS), alpine meadow (AM) and the alpine meadow steppe (AMS) were studied in Naqu Prefecture of northeastern Tibet. The landscapes are at about 4500 m average altitude and have about −0.9 to −3.3 °C mean annual temperature. There is about 48–51% annual relative humidity and about 380 mm mean annual precipitation, while the rain occurs mainly during April to September and it is short and cooling [17]. The dominant species are *Carex moorcroflii*, *Poa annun*, *Kobresia humilis*, *Stipa purpurea* and *Artemisia argyi*.

The samples of alpine desert steppe (ADS) and alpine desert (AD) were studied in Ngari Prefecture, where the mean altitude is about 4500 m while there is about 0 °C mean annual temperature and about 180 mm mean annual precipitation [18]. The dominant species are *Stipa capillata Linn.*, *Suaeda corniculate* and *Artemisia wellbyi*.

No specific permissions were required for research locations and the studies did not involve endangered or protected species. All the authors consented for publication.

### 2.2. Experimental Samples

In August 2016, 21 sites were selected in the northern Tibetan Plateau. The geographical locations, sample sites, numbers, important value (IV) and soil water content (SWC) are shown in Figure 1 and Table 1. In each site, the AGB and BGB under three blocks of 1.0 m × 1.0 m was harvested. We collected BGB at 0–15 cm of depth, where the most BGB was found [19]. AGB and BGB samples were dried at 80 °C for about 24 h, then a balance was used to measure the weight. Total biomass (TB) was calculated by AGB plus BGB. Root:shoot ratio (R/S) was calculated by using BGB divided by AGB. The top soil layers were also sampled in 0–15 cm.

### 2.3. Methods

#### 2.3.1. Importance Value (IV)

The calculation formula is expressed as follows:(1)$$IV=\frac{D+C+F}{3}$$
where *D* is the relative density (percentage of numbers of one species to the total numbers of all species in a quadrat), *C* is the relative cover rate (percentage of the coverage of one species to that of all species in a quadrat), and *F* is the relative frequency (percentage of the frequency of one species to the sum of frequencies of all species in the quadrats).

#### 2.3.2. Biomass Allometric Method

According to the allometric biomass partitioning theory [20,21], the AGB and BGB partitioning pattern can be described using Equation (2):*M_A_* = *βM_B_^α^* or *log M_A_* = *αlogM_B_* + *log^β^*(2)
where *M_A_* and *M_B_* represent the AGB and BGB, respectively; where *β* is the allometric constant, *α* is the scaling exponent [22].

#### 2.3.3. Biomass and Soil Analysis Methods

All the soil and biomass samples were ground by a mill (Spex Sample Prep 8000D, Metuchen, NJ, USA). The method of wet oxidation by K_2_Cr_2_O_7_-H_2_SO_4_ was used to measure the total organic carbon concentration (SOC) of soil [23]. The plant and soil samples were digested using the H_2_SO_4_-H_2_O_2_ digestion procedure, and then the total nitrogen (N) concentrations were measured using the FLOWSYS Ⅲ system (Flowsys, Systea, Italy). The biomass C, N stocks were determined using the biomass multiplied with the respective total C, N concentrations [24]. The soil water content (SWC) was also measured using the oven-drying weight method. First, weigh the fresh soil weight (FW), and then, put the soil in the 105 °C oven to dry to the constant weight (DW). SWC (%) = (FW − DW)/FW × 100%.

#### 2.3.4. Intrinsic Water Use Efficiency (WUEi) Calculation

AGB was randomly collected and mixed together as a block, forming a composite sample and then the carbon isotope composition (*δ^13^Cp*) was measured. An Iso Prime continuous-flow stable isotope ratio mass spectrometer (GV Instruments Ltd., Manchester, UK) was used to analyze the stable carbon isotope composition of all leaf sample powders. According to the equation (2) of Farquhar et al. [25], assuming an isotope composition in ambient air as −8.0‰, the leaf carbon isotope discrimination (Δ) was calculated:Δ = [(*δ^13^Ca* − *δ^13^Cp*)/(1 + *δ^13^Cp*)] × 1000(3)
where *δ^13^Ca* is the isotope composition of atmospheric CO_2_ and *δ^13^Cp* is the isotope composition of plant samples.

Previous studies had reported that Δ had a relationship with the ratio of intercellular CO_2_ concentration (*Ci*) to that in ambient atmosphere (*Ca*) (*Ci/Ca*). Equation (4) expresses this relationship, where *a* was the fractionation of CO_2_ in air (4.4%) and *b* was the fractionation associated with CO_2_ fixation (27%) by plants [26]: (4)$$\Delta =a+(b-a)\frac{Ci}{Ca}$$

At the leaf level, WUEi is defined as the ratio of net photosynthetic rate (*A*) to stomatal conductance (*gs*) (Equation (5)), where the ratio of diffusivities of water vapor to CO_2_ in the atmosphere is 1.6.
(5)$$\mathrm{WUEi}=\frac{A}{gs}=\frac{\left(Ca-Ci\right)}{1.6}=Ca(1-\frac{Ci}{Ca})/1.6$$

During the growing season in 2016, the *Ca* was 400 ppm in Tibetan Plateau [27]. Thus, according to Equations (3)–(5), the WUEi can be calculated from *δ^13^Cp*.

#### 2.3.5. Data Statistical Analysis

The differences of all the data were computed using one-way ANOVA in SPSS (SPSS software version 22.0, Chicago, IL, USA). The Tukey’s HSD test was used to compare the differences between means. Based on a covariance matrix, using the package CANOCO 5.0 (Microcomputer Power, Inc., Ithaca, NY, USA), redundancy analysis (RDA) was conducted to evaluate the relationships between biomass, WUE, C, N concentrations and SWC parameters under different alpine grassland types. The Spearman’s correlation test function of SPSS is used to analyze the correlations between different parameters.

## 3. Results

### 3.1. Biomass

#### 3.1.1. Biomass Characteristics

Differences in AGB, BGB, TB and R/S between the five alpine grasslands are shown in Table 2. The AGB was 24.209 g/m^2^ in AM and it was the highest and almost twice that of AD. In AM, the BGB was seven times greater than that in AD. Moreover, the highest R/S was in AM, the lowest was in AD.

There was significant positive correlation between BGB and the TB; R/S (*p* < 0.01) and the TB also had significant correlation with R/S (*p* < 0.01).

#### 3.1.2. Biomass Partitioning

In Figure 2, according to the allometric relationship from the BGB, the empirical estimates of AGB were obtained. As expected, the AGB and BGB partitioning results significantly satisfied the allometric biomass partitioning theory which showed a power function. For the different types of alpine grassland, the curves and equations were all different. The *α* value in AM was close to 1, which was 0.955. The lowest *α* value was in AMS, 0.273. For the alpine grassland model, the F test (F = 222.596, *p* < 0.01) indicated that this model gave estimates without systematic error.

### 3.2. Biomass Organic Carbon (C), Nitrogen (N) Concentrations and Stocks

The C, N concentrations in AGB, BGB and C-N ratios (C/N) in AGB showed significant differences between grassland types (Table 3). The highest C concentration (448.892 mg/g) was obtained under AGB in AM. The lowest C concentration (303.944 mg/g) was also obtained under AGB in AD. The N concentrations of AGB were higher than that in BGB. Moreover, the highest C/N (39.616) was obtained in AD under BGB and the lowest C/N (22.420) was also obtained under AGB in AD.

The C, N concentrations and C/N of AGB and BGB exhibited large variations between study sites, ranging from 264 to 495 mg/g for AGB C concentration, 249–485 mg/g for BGB C concentration, 12.03–19.52 mg/g for AGB N concentration, 6.51–13.9 mg/g for BGB N concentration, 20.178–35.102 for AGB C/N and 23.943–48.233 for BGB C/N (Figure 3).

There were significant grassland type differences in AGB, BGB C stocks, AGB, BGB N stocks and TB C, N stocks (Table 4). The highest BGB C stocks were in AM which was almost seven times greater than the AGB C stocks. The lowest total vegetation C storage was in AD, 18.743 g C/m^2^. The BGB N stocks in AM was about six times greater than the AGB N stocks. The N storage of vegetation ranged from 0.604 to 3.387 g N/m^2^ and the highest was in AM. The C, N storage of BGB accounted for 81.9% and 78.4%, respectively.

### 3.3. Soil Water Content (SWC) and Vegetation Water Use Efficiency (WUE)

There were significant differences in SWC under 5 alpine grasslands (F = 19.240, *p* < 0.01) and the highest value was in AM, 13.737%, while the lowest was in AD, only 2.612%. There were significant differences in *δ^13^C* and WUEi under 5 alpine grassland types (F = 125.424, *p* < 0.01) while the highest mean of foliar *δ^13^C* and WUEi was in AD (−24.987‰, 105.960 μmol CO_2_/H_2_O, respectively) (Figure 4) and the lowest mean was in AM (−27.489‰, 76.984 μmolCO_2_/H_2_O, respectively).

### 3.4. The Correlation among Biomass, C, N Concentrations, SWC and WUEi

Based on the redundancy analysis (RDA), correlations between biomass, C, N concentrations, SWC and WUEi are shown in Figure 5. All the axes of the RDA (pseudo-F = 8.7, *p* = 0.002) including the first axis (pseudo-F = 44.4, *p* = 0.002) were significant. Most indices of biomass (AGB, BGB and R/S), and C, N concentrations (AGB C, BGB C, AGB N and BGB N) together with SWC were related to this axis, which means these parameters all had significant positive correlation with each other. In the opposite direction, there was an interrelationship of WUEi, TB and BGB C/N to the opposite axis, which means that these parameters had significant negative correlations with those parameters mentioned above. 

Significant negative correlations were found between WUEi and C, N concentrations, C/N in AGB and SWC (*p* < 0.05), while the correlation with C/N of BGB was not significant (Table 5). AGB was significantly positively correlated with C, N concentrations of AGB, N concentrations of BGB and SWC (*p* < 0.01). Additionally, BGB had significant positive correlations with C, N concentrations of AGB, BGB and SWC (*p* < 0.01), while it had significant negative correlations with C/N of BGB (*p* < 0.05). For R/S, the relationships among them were similar to BGB (*p* < 0.05). For TB, it was significantly positively correlated with C, N concentrations of AGB, BGB and SWC (*p* < 0.05).

## 4. Discussion

### 4.1. Biomass

Biomass from the vegetation component of grasslands is an important index for ecosystem functioning. Different plant species compete mainly for nutrients and water in unproductive grasslands which forms different dominant species under varied environments. In AM, the highest IV of species was *Carex* spp. and in AD, it was *Stipa capillata* Linn. (Table 1). According to the research of Lu et al., the biomass of different species varied greatly in the fragile environment and the *Carex* spp. always had higher AGB than the *Stipa capillata* Linn. [28]. Our research had similar results, that in AM, the biomass (AGB, BGB, TB) was greater than that in AD (Table 2).

In order to match the prerequisites of the physiological activities and functions of these organs, the plants have to balance the allocation of leaves and roots in a certain way. The allometric biomass partitioning theory [29] is always used to examine plant growth under an available resource. Plants always respond to some relative shortage of any essential resources by increasing their allocations to the functions and structures, which also have responsibility for the acquisition of those limiting resources or by decreasing the loss of the limiting resources [30]. Allometric biomass partitioning theory also shows a series of scaling relationships that reflect how plants should partition biomass based on the constraints of maximizing photosynthetic harvesting capacity and resource transport under limiting resource environments [31]. The results of these grassland difference studies suggested a significant effect of type on biomass allometry and partitioning (Table 2 and Figure 2). In Equation (2), the parameter *α* reflects the relative growth ratio between AGB and BGB. When it equals 1, it means that the growth of each part is relatively in the same proportion and the plant is growing in a steady state. However, most times, it will deviate from unity because of the consequence of interspecific variation, ontogenetic development differences or environmental perturbation such that *α* value is changed [32]. For our results, in AM, the *α* was close to 1 which means the environment was steadier than other grassland types (Figure 2A). In AMS and AD, the *α* was far from 1 which means these two grassland types of environment had soil water limiting (SWC) which allocated more to one part of the plant and then caused inconsistency of AGB and BGB (Figure 2C,E, Table 1). Some previous studies showed that in the AD habitat, fragmentation caused environment unsteadiness that was more severe than that in AM [33,34].

Previous studies showed that if the plants are grown in a low nutrient supply or low water conditions, they will show an increased allocation to the roots [35]. Our research also had similar results. For all the grassland ecosystems, the BGB was all higher than the AGB because of the stress condition in the Northern Tibetan Plateau. So, for future research about this, when we want to know if the plants actively alter their assimilation partitioning, the study also needs to determine whether the allocation differences exist and persist when adjusting to a given environment. Moreover, we also need to know that the mechanism which controlled the partitioning was not only one gene, but also all organs relating to relative source and sink strengths [36].

### 4.2. C and N Stocks

The impact of different land cover on biogeochemical cycles, particularly the C and N cycles, has been the subject of much attention in recent years [37]. Quantification of the differences in C pool and N nutrient is fundamental to the understanding of the effects of different land use on ecosystem function [38]. Although, under per unit area, the grassland vegetation represented an insignificant pool of C and N compared to the tree biomass pool [39]. However, for the abundant species compositions and large growing areas of grassland, it can also make great contributions to the global C and N pools. In our study sites, there were lots of species, higher abundance and larger growing areas which contributed to the fixation of C, N pools. Vegetation C, N storage can depend on soil C pool and N availability while the dependence is in both ways. The soil C and N pools also generally depend on litterfall from the vegetation [40]. Our results supported the hypotheses because the higher soil C, N concentrations occurred in AM, AS and ADS and they had higher C, N storage of vegetation (Table 3). In addition, a higher root biomass (BGB) and R/S contributed to plant N uptake (AGB N and BGB N) and increased the soil N availability (Table 5 and Figure 5). Moreover, larger BGB also can contribute more C and N to soil in the mixed-grass and semi-arid grassland of the northern area, respectively [41,42], which was the same as our results (Table 2 and Table 3). For our results, although high C, N storage in BGB was similar, the effects of grassland type on this pattern were different [43] which suggested that grassland type changed the aboveground and belowground allocation of C, N within the plant. Belowground C, N allocation may allow compensatory responses to different conditions [44] and might also facilitate the recovery of vegetation after some other natural disturbance [45]. The quantification of the ecosystem C, N pools of grasslands is important, especially the amount of C, N stored, while that in belowground was usually greater than that in aboveground (Table 4).

### 4.3. Water Use Efficiency

In many types of grassland ecosystems, soil water is variable and it can often be the most limiting to plant growth [46]. In our study, the SWC in AM was six times greater than that in AD (Table 1), which limited the AD biomass significantly, while the TB in AM was about five times more than that in AD. Therefore, the WUE, so sensitive to water limiting, is a crucial index used in the northern Tibetan Plateau. Stable carbon isotope as an expression of WUE has often been used to explain some crucial interdependence of the carbon and water relations of plants [47]. The *δ^13^C* and the discrimination of community (Δ) is suggested as a sensitive long-term monitoring of plant physiological changes [48], and the WUEi (at the leaf level, which refers to the water used per unit carbon gain) was derived from plant *Δ* (Equations 3–5). When a plant faces soil water limiting conditions, a strong increase is shown in VPD and plants close their stomata to reduce transpiration, eventually leading to a lower *Ci* [49]. With the stronger stomatal limitation of photosynthesis, the leaves or needles will exhibit a low *Ci/Ca* and a reduced discrimination, along with an improved WUEi [50]. In our research condition of AD, the water was limited; it had the results that the WUEi was increased while the Δ was increased in AD (Figure 4) which may be explained by the fact that SWC was limiting (Table 5 and Figure 5).

Under a stressful environment, vegetation *Δ* related to WUEi is changed during leaf gas diffusion and Rubisco-dependent reactions [51]. For the former studies, under limiting resources, WUEi is negatively correlated with plant dry mass accumulation [25]. Our research had the similar results in that there were significant negative correlations between AGB, BGB and WUEi (Figure 5). When plants are under a water limiting environment, stomatal conductance is at a low level to reduce water loss and thus increases WUEi; however, low stomatal conductance also constrains carbon gain [52]. Thus, in the present study, the relationships between WUEi and AGB C, BGB C were negative (Table 5). Higher AGB and BGB N concentrations usually correlate with higher rates of photosynthesis from stomata, because most of N is associated with photosynthetic proteins. If the higher WUEi is maintained, N synthesis is reduced because of the reduced rate of photosynthesis after the stomata closed [53]. Our research confirmed the results that there were negative correlations between WUEi and AGB N, BGB N.

## 5. Conclusions

The northern Tibetan Plateau is a fragile ecosystem, where alpine grasslands are the major types of pasturelands; the growth plants’ situation, condition, relationships and differences among them are our main research scopes. In our research, we studied the biomass partitioning, C, N stocks, soil property and WUE differences of five types of alpine grasslands in the northern Tibetan Plateau area. There were significant differences among five alpine grasslands of growth indices (AGB, BGB, TB and R/S), the highest value of which was shown in AM. The plants of AM were in a steadier environment according to allometric theory. The highest C, N stocks of belowground biomass were in AM. There was significant difference in *δ^13^C* and WUEi under five alpine grassland types while the highest means of foliar *δ^13^C* and WUEi were in AD. According to our research results, it is proposed that the AM grassland type may be actively growing well and could use limited water to produce more biomass there. However, in the future, more seasonal and interannual research would need to be surveyed to evaluate the results and some other relative meteorological indexes also need to be surveyed.

## Figures and Tables

**Figure 1 ijerph-19-13026-f001:**
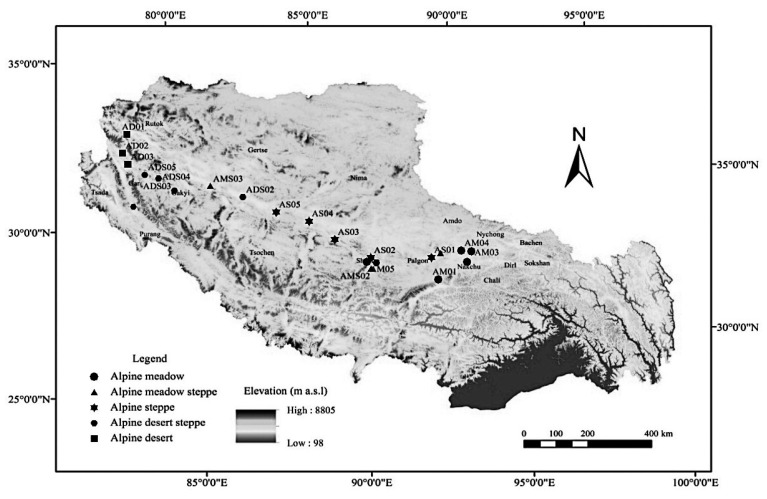
Research areas and sampling sites.

**Figure 2 ijerph-19-13026-f002:**
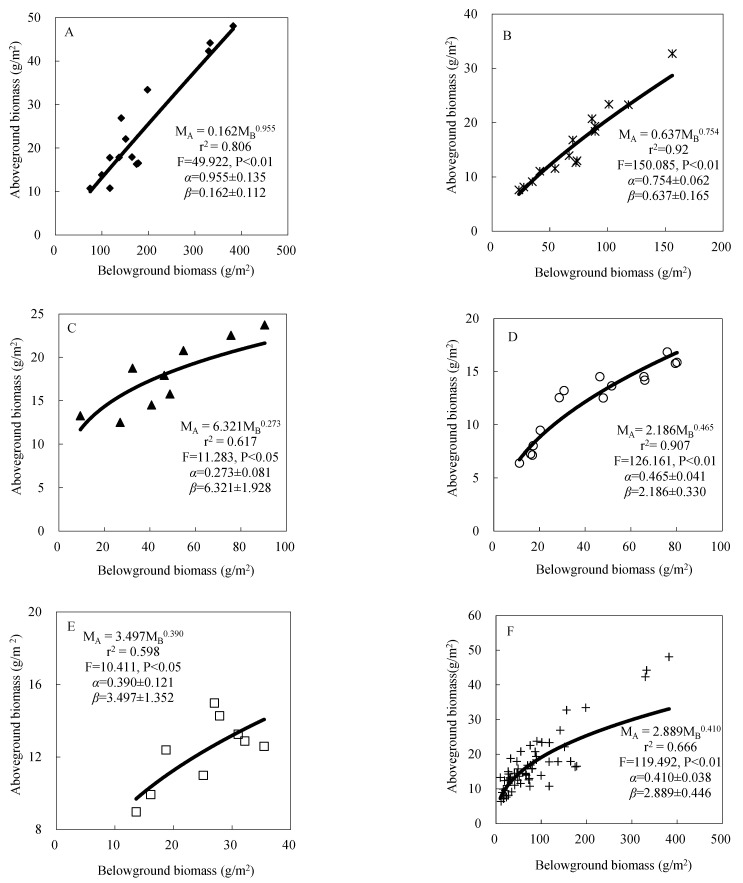
The Empirical regression model between belowground biomass (BGB) and aboveground biomass (AGB) in 5 alpine grassland types. The *α* and *β* values presented in each figure were mean ± standard errors. (**A**): alpine meadow (AM), (**B**): alpine steppe (AS), (**C**): alpine meadow steppe (AMS), (**D**): alpine desert steppe (ADS), (**E**): alpine desert (AD) and (**F**): alpine grassland types.

**Figure 3 ijerph-19-13026-f003:**
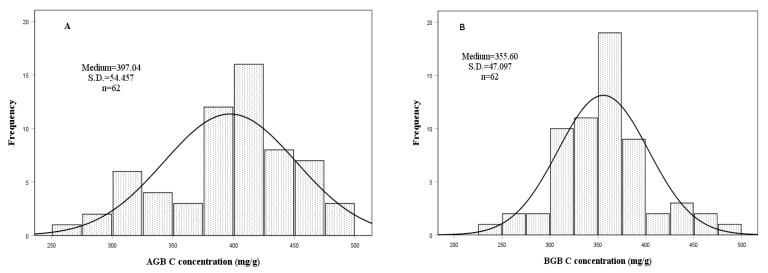
The frequency distribution of C, N concentrations and C/N of belowground biomass (BGB) and aboveground biomass (AGB) in 5 alpine grassland types. (**A**): alpine meadow (AM), (**B**): alpine steppe (AS), (**C**): alpine meadow steppe (AMS), (**D**): alpine desert steppe (ADS), (**E**): alpine desert (AD) and (**F**): alpine grassland types.

**Figure 4 ijerph-19-13026-f004:**
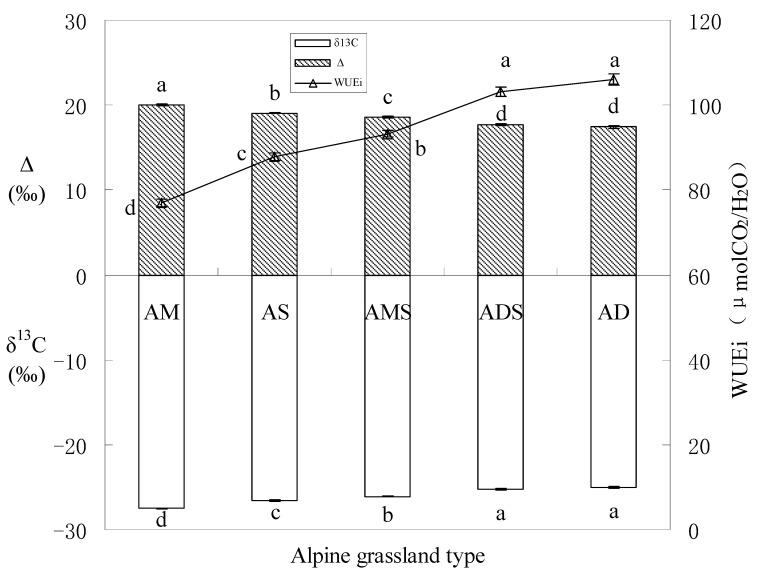
The *δ^13^C*, Carbon isotope discrimination (Δ) and intrinsic water use efficiency (WUEi) of aboveground biomass (AGB) in 5 alpine grassland types. Letters are reported only for significant differences in one-way ANOVA post hoc Tukey’s HSD.

**Figure 5 ijerph-19-13026-f005:**
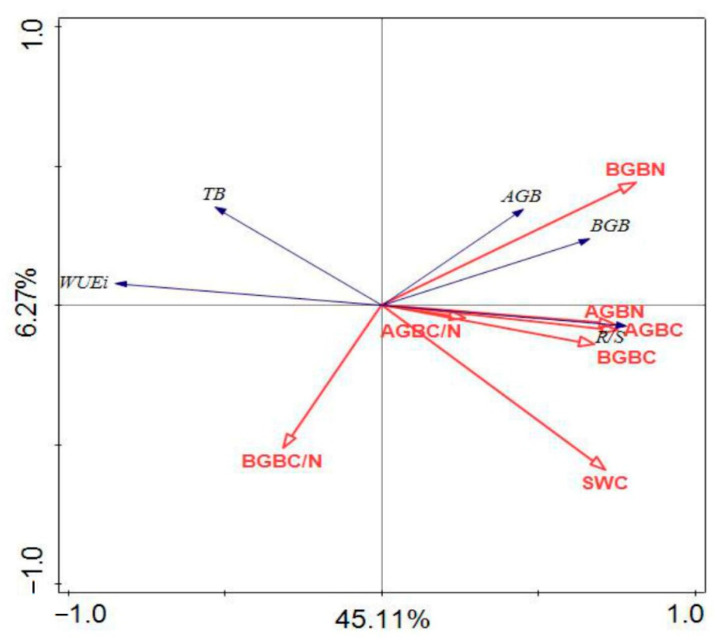
Redundancy analysis (RDA) of biomass, WUEi and C, N concentration, soil water content in 5 alpine grassland types. Aboveground biomass (AGB), Belowground biomass (BGB), total biomass (TB), BGB to AGB ratio (R/S), carbon concentration of AGB (AGBC), carbon concentration of BGB (BGBC), nitrogen concentration of AGB (AGBN), nitrogen concentration of BGB (BGBN), C to N ratio of AGB (AGBC/N), C to N ratio of BGB (BGBC/N), soil water content (SWC) and intrinsic water use efficiency (WUEi).

**Table 1 ijerph-19-13026-t001:** Sample geographical locations and other information of the five alpine grasslands.

Grasslands Type	Site	Latitude N	Longitude E	Elevation	Dominant Species	Important Value (IV)	Soil Water Content (SWC, %)
alpine meadow (AM)	Nam Co	91.112	30.750	4812	*Carex* spp.	0.257	8.079
	Naqu	91.980	31.377	4594	*Poa pratensis*	0.217	19.467
	Nima	92.070	31.729	4670	*Kobresia humilis*	0.174	13.666
	Anduo	91.730	31.716	4655	*Kobresia humilis*	0.248	14.737
	Shenzha	88.699	30.957	4654	*Edelweiss*	0.168	12.737
alpine steppe (AS)	Bange	90.777	31.389	4619	*Carex* spp.	0.27	13.007
	Shenzha	88.700	31.124	4735	*Potentilla chinensis*	0.228	10.106
	Nima	87.483	31.505	4648	*Stipa capillata Linn.*	0.277	12.105
	Nima	86.503	31.932	4718	*Carex* spp.	0.238	11.106
	Gaize	85.356	32.032	4785	*Stipa capillata Linn.*	0.19	9.206
alpine meadow steppe (AMS)	Bange	91.058	31.552	4544	*Stipa capillata Linn.*	0.309	10.980
	Shenzha	88.702	30.957	4664	*Carex* spp.	0.216	10.106
	Gaize	82.978	32.429	4421	*Artemisia argyi*	0.315	11.897
alpine desert steppe (ADS)	Shenzha	88.712	30.960	4733	*Carex spp.*	0.257	8.765
	Nima	84.133	32.291	4438	*Stipa capillata Linn.*	0.476	12.876
	Gaize	81.826	32.071	4606	*Ptilotricum wageri*	0.154	8.986
	Gaize	81.206	32.327	4543	*Potentilla chinensis*	0.153	14.176
	Ge’gyai	80.715	32.347	4628	*Oxytropis ochrocephala*	0.148	9.162
alpine desert (AD)	Rutog	79.755	33.432	4266	*Suaeda corniculate*	0.358	2.201
	Rutog	79.784	32.838	4443	*Stipa capillata Linn.*	0.391	3.562
	Rutog	80.061	32.544	4400	*Artemisia macilenta*	0.153	2.074

**Table 2 ijerph-19-13026-t002:** Differences of aboveground biomass (AGB), belowground biomass (BGB), total biomass (TB) and BGB to AGB ratio (R/S) ratio in 5 alpine grassland types.

Grassland type	AGB (g/m^2^)	BGB (g/m^2^)	TB (g/m^2^)	R/S
AM	24.209 ± 3.402a	185.760 ± 25.283a	209.969 ± 28.490a	7.952 ± 0.481a
AS	16.127 ± 1.793b	73.606 ± 9.217b	89.733 ± 10.966b	4.451 ± 0.198b
AMS	17.767 ± 1.347b	47.274 ± 8.196c	65.042 ± 9.397bc	2.544 ± 0.038cd
ADS	12.133 ± 0.912b	43.644 ± 6.513c	55.777 ± 7.358bc	3.322 ± 0.312c
AD	12.251 ± 0.654b	25.234 ± 2.520c	37.485 ± 3.010c	2.039 ± 0.156d
F Value	5.921 **	20.515 **	18.319 **	49.423 *

Values are represented as mean ± standard error, and letters are reported only for significant differences in one-way ANOVA post hoc Tukey’s HSD. * *p* < 0.05, ** *p* < 0.01.

**Table 3 ijerph-19-13026-t003:** Differences of above-ground biomass (AGB) and soil C, N concentrations, below-ground biomass (BGB) and soil C, N concentrations, AGB C/N and BGB C/N ratios in 5 alpine grassland types.

Grass-land Type	AGB C Concentra-tion (mg/g)	BGB C Concentra-tion (mg/g)	AGB N Concentra-tion (mg/g)	BGB N Concentra-tion (mg/g)	AGB C/N	BGB C/N	Soil N Concentra-tion (mg/g)	Soil C Concentra-tion (mg/g)
AM	448.892 ± 7.784a	403.036 ± 11.778a	17.511 ± 0.204a	12.306 ± 0.208a	25.630 ± 0.302a	32.911 ± 1.185b	1.819 ± 0.179a	32.934 ± 1.466a
AS	413.467 ± 5.530b	366.600 ± 8.900b	17.015 ± 0.312a	10.310 ± 0.340b	24.479 ± 0.570abc	35.952 ± 1.169ab	0.855 ± 0.054b	15.150 ± 0.120b
AMS	423.056 ± 9.950b	324.056 ± 13.379c	17.003 ± 0.187a	10.098 ± 0.375b	24.916 ± 0.708ab	32.566 ± 2.037b	0.442 ± 0.077c	7.437 ± 0.229c
ADS	372.467 ± 9.214c	330.633 ± 9.990c	15.748 ± 0.347b	9.783 ± 0.412b	23.848 ± 0.923bc	34.567 ± 1.657b	0.170 ± 0.001d	4.035 ± 0.051d
AD	303.944 ± 7.122d	336.667 ± 5.891bc	13.559 ± 0.147c	8.651 ± 0.478c	22.420 ± 0.495c	39.616 ± 1.710a	0.086 ± 0.016e	1.818 ± 0.151e
F Value	47.715 **	9.714 **	26.059 **	12.828 **	2.780 *	2.821 *	43.089 **	262.484 **

Values are represented as mean ± standard error, and letters are reported only for significant differences in one-way ANOVA post hoc Tukey’s HSD. * *p* < 0.05, ** *p* < 0.01.

**Table 4 ijerph-19-13026-t004:** Nitrogen (N), carbon (C) stocks of aboveground biomass (AGB), belowground biomass (BGB) and total biomass (TB) in 5 alpine grassland types.

Grassland Type	AGB C Stock (g C/m^2^)	BGB C Stock (g C/m^2^)	AGB N Stock (g N/m^2^)	BGB N Stock (g N/m^2^)	TB C Stock (g C/m^2^)	TB N Stock (g N/m^2^)
AM	12.104 ± 1.701a	92.880 ± 12.642a	0.472 ± 0.067a	2.915 ± 0.472a	104.984 ± 14.245a	3.387 ± 0.534b
AS	8.064 ± 0.896b	36.803 ± 4.609b	0.338 ± 0.043ab	1.063 ± 0.161b	44.867 ± 5.483b	1.400 ± 0.203b
AMS	8.884 ± 0.674b	23.637 ± 4.098bc	0.360 ± 0.030ab	0.768 ± 0.175b	32.521 ± 4.699bc	1.128 ± 0.200b
ADS	6.067 ± 0.456b	21.822 ± 3.256bc	0.259 ± 0.022b	0.790 ± 0.127b	27.889 ± 3.679bc	0.959 ± 0.146b
AD	6.126 ± 0.327b	12.617 ± 1.260c	0.276 ± 0.019b	0.328 ± 0.037b	18.743 ± 1.505c	0.604 ± 0.055b
F Value	5.921 **	20.515 **	3.894 **	14.795 **	18.319 **	13.058 **

Values are represented as mean ± standard error, and letters are reported only for significant differences in one-way ANOVA post hoc Tukey’s HSD. ** *p* < 0.01.

**Table 5 ijerph-19-13026-t005:** Correlation among aboveground biomass (AGB), belowground biomass (BGB), total biomass (TB), BGB to AGB ratio (R/S), carbon (C), nitrogen (N) concentrations, soil water content (SWC) and intrinsic water use efficiency (WUEi).

Index	WUEi	AGB	BGB	TB	R/S
AGB C	−0.710 **	0.354 **	0.448 **	0.443 **	0.504 **
BGB C	−0.592 **	0.247	0.441 **	0.426 **	0.518 **
AGB N	−0.693 **	0.359 **	0.436 **	0.433 **	0.522 **
BGB N	−0.632 **	0.508 **	0.638 **	0.632 **	0.643 **
AGB C/N	−0.259 *	0.119	0.167	0.163	0.163
BGB C/N	0.207	−0.325 *	−0.298	0.162	−0.265 *
SWC	−0.626 **	0.103	0.332 **	0.312 *	0.656 **

* *p* < 0.05, ** *p* < 0.01.

## Data Availability

The data and materials are available in the manuscript.

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
