# Peer review of "Biomass, Carbon and Nitrogen Partitioning and Water Use Efficiency Differences of Five Types of Alpine Grasslands in the Northern Tibetan Plateau"

_ijerph, 2022, doi:10.3390/ijerph192013026_

Round 1

Reviewer 1 Report

The paper studied the biomass, carbon and nitrogen partitioning and water use effi- 2ciency differences of five types of alpine grasslands in the 3Northern Tibetan Platea. There are many shortcomings and need to improved.

1.     The language should be improved by a professional English speaker;

2.     Line 23: Pleased give the full name of WUEiï¼›

3.     Line 32: change km2 to km2ï¼›

4.     Line 35-37: Please simply the sentence, I can’t understand the meaning of it;

5.     Line 44: I can’t understand why there is a semicolon;

6.     Line 55-58: The research progress was too short;

7.     Change P to P;

8.     The figure 1 and figure 3 was not clear;

9.     The conclusion was too short, and lots of sentence was focused on the background,

10.  Line 138: Which index is 24.209g/m2ï¼›

Reviewer 2 Report

The authors study the water use efficiency of five different alpine grassland types in the Northern Tibetan Plateau. An objective was to determine how biomass, water use efficiency, soil carbon and nitrogen concentrations, and soil water content relate to each other. The authors show significant differences in aboveground, belowground, and total biomass and in root/shoot ratios. Alpine desert (AD) showed the least discrimination against carbon-13 and thus the highest water use efficiency.

Comments

The other differences among the grassland types and the relationships of aboveground and belowground biomass, nutrient stocks and water use efficiency all seem reasonable. I just have a few questions.

In Table 1, how was “Importance value’ calculated?

Lines 187-188.  It is stated here that total biomass (TB) ‘had significant negative correlation with those parameters mentioned above’, which include AGB and BGB. That does not seem reasonable to me, and Table 5 shows a positive correlation.  Maybe more explanation of the redundancy analysis results are needed.

Lines 288-289.  I am not sure what the authors mean by ‘C, N availability’.  N availability is certainly important to plants, but I suppose that soil C is just a measure of the organic content of soil, which has a benefit, for example, in holding moisture.

Line 274. I would like to see more explanation of why a value of alpha (in Equation 1) close to 1 for AM ‘means the environment was steadier that other grassland types’. I am not sure what ‘steadier’ means in this context. I would assume that alpha close to 1 means that AGB and BGB are linearly related, which perhaps means that water or nutrients are not limiting.

Although the manuscript is understandable, the English grammar is poor and must be improved. I have made some suggested changes below, but the manuscript should be looked over carefully by a native English speaker to make further improvements.

Line 11. Change ‘is covered most area’ ‘covers most areas’

Line 15. Change ‘of WUE five kinds of vegetation’ to ‘and WUE of five kinds of vegetation’

Line 18. Change ‘highest value of different index was showed’ to ‘highest values of the different indices were shown’

Line 21. Change ‘was in’ to ‘were in’

Line 23. I think WUEi should be defined as ‘intrinsic water use efficiency’ here

Line 24. Change ‘was in AD’ to ‘were in AD’

Line 28. Change ‘suggested’ to ‘it is suggested’ and change ‘AS may be an active grassland type’ to ‘AS may be an actively growing grassland type.

Line 33. Rewrite as “Grasslands are an important vegetation type of about 50 million km2, covering about 40% of the earth’s terrestrial area”.

Line 36. Change ‘in recent years which is also facing lots of environmental degradations’ to ‘and in recent years is also facing much environmental degradation’

Line 41. Change ‘which is including’ to ‘which includes’

Line 42. Change ‘subsequent resulting in the implications of C and N cycles’ to ‘consequently resulting in implications for C and N cycles’

Line 49.  Insert comma after ‘plants’

Line 50.  Change ‘was the critical’ to ‘were the critical’

Line 51. Change ‘during’ to ‘along’

Line 54. Change ‘which is also’ to ‘is also’

Line 56. Change ‘had none’ to ‘had no’ and change ‘none-grazing’ to ‘non-grazing’

Line 59. Change ‘mainly’ to ‘main’

Lines 60-61.  Change to ‘whether the differences in the grassland types are related to WUE differences in the Northern Tibetan Plateau’

Line 69. Change ‘have about’ to ‘are at about’

Line 72. Change ‘raining’ to ‘rain’

Line 82. Change ‘were showed’ to ‘are shown’

Line 91. Change ‘the Equation (1)’ to ‘Equation (1)’

Line 92, Equation (1). Alpha should be an exponent of MB

Line 100.  Change ‘are measured’ to ‘were measured’

Line 102. Change ‘concentration’ to ‘concentrations’

Line 105. Change ‘and then measured the carbon isotope composition (delta13Cp)’ to ‘and then the carbon isotope composition (delta13Cp) was measured’

Line 111. Change ‘Where’ to ‘where’

Line 113. Change ‘had relationship’ to ‘has a relationship

Line 114. Change ‘expressed’ to ‘expresses’

Line 120. Change ‘was 1.6’ to ‘is 1.6’

Line 125. I think ‘were used’ should be changed to ‘were computed using’

Line 126. Change ‘test used’ to ‘test was used’

Line 137. Change ‘was shown’ to ‘is shown’

Line 138. Change ‘it were’ to ‘it was’

Line 146. Change ‘estimating AGB were obtained’ to ‘estimates of AGB were obtained’

Line 150. Change ‘according the F test’ to ‘the F test’

Line 151. Change ‘indicating’ to ‘indicated’

Line 154. Change ‘showed significant grassland type differences’ to ‘showed significant differences between grassland types’

Line 168. Change ‘seven times than’ to ‘seven times greater than’

Line 169. Change ‘six times than’ to six times greater than’

Line 174. Change ‘It had significant difference of SWC’ to ‘There were significant differences in SWC’

Line 176. Change ‘There was significant difference’ to ‘There were significant differences’

Line 181. Change ‘correlation’ to ‘correlations’

Line 182. Change ‘were showed’ to ‘were shown’

Line 183. Change ‘were all significant’ to ‘were significant’

Line 184. Change ‘index’ to ‘indices’

Line 185. Change ‘which mean’ to ‘which means’

Line 188. Change ‘correlation’ to ‘correlations’

Line 196. Change ‘it significantly’ to ‘it was significantly’

Line 263.  I think that ‘prerequisites’ may be better than ‘preformation’ here

Line 267. Change ‘responsible’ to ‘responsibility’

Line 270. Maybe change ‘the equation’ to ‘Equation (1)’ and change ‘which means the plant growing’ to ‘it means that the plant is growing’

Line 272. Change ‘while the’ to ‘such that’

Line 275. Change ‘mean’ to ‘means’

Line 278. Change ‘plants grown’ to ‘plants are grown’ and change ‘less’ to ‘low’

Line 279. Change ‘which will show’ to ‘they will show’

Line 279. Change ‘In our research, it also had similar results during all the grassland ecosystems’ to ‘Our research also had similar results. For all the grassland ecosystems’

Line 283. Change ‘do’ to ‘determine’

Line 288.  Change ‘Vegetations’ to ‘Vegetation’

Line 290.  Change ‘of vegetations’ to ‘in vegetation’

Line 291. Change ‘was contribution’ to ‘contributed to’

Line 300. Insert comma after ‘important’

Line 300. Change ‘while it was in the belowground was usually greater than that in the aboveground (Table 4)’ to ‘while that belowground was usually greater than that aboveground (Table 4)’

Line 304.  Change ‘limiting plant’ to ‘limiting to plant’ and change ‘six time than’ to ‘six times greater than’

Line 306. Change ‘times than’ to ‘times greater than’

Line 306. Change ‘always used’ to ‘has often been used’

Line 310. Change ‘for saving water the mechanism’ to ‘in order to save water, the mechanisms’

Line 311. Change ‘which referred to’ to ‘which refers to’

Line 315. Change ‘which may explained as the SWC and nutrient limiting’ to ‘which may be explained by the fact than SWC and nutrient are limiting’

Line 318. Delete comma after ‘Plateau’

Line 320. Change ‘differences are the maim research scopes’ to ‘differences among them are our main research scope’

Line 323. Change ‘actively growth well’ to ‘actively growing well’

Line 324. Change ‘in there’ to ‘there’

Line 326. Change ‘would be surveyed’ to ‘would need to be surveyed’

Reviewer 3 Report

Dear Authors,

I read the paper "Differences in biomass, carbon and nitrogen partitioning, and water use efficiency of five types of alpine grasslands in the Northern Tibetan Plateau."

It's an intriguing topic, but I believe English editing is required because the reading is difficult for reviewers.

Concentrate on the differences between sites. Furthermore, the discussion is extremely poor.

Please use these suggestions to improve the paper.

Round 2

Reviewer 2 Report

The authors study the water use efficiency of five different alpine grassland types in the Northern Tibetan Plateau. The show significant differences in aboveground, belowground, and total biomass and in root/shoot ratios. Alpine desert (AD) showed the least discrimination against carbon-13 and thus the highest water use efficiency.

Comments

 I reviewed an earlier version of this manuscript. The authors have responded to my comments.  I have a few more comments on the revised version.

 As I requested in my review of the original version, the Importance value (IV) has been defined. However, the authors should explain in more detail what the parameters D, C, and F mean.  Now they are just defined a ‘relative density’, ‘relative cover rate’ and ‘relative frequency’, but more details are needed.

 Lines 339. “Although the grassland vegetation represented an insignificant pool of C and N compared to total ecosystem pools, and even to the tree biomass pool [39]. However, C, N, pools in grassland vegetation can be expected to be higher since they depend on species composition, abundance and large growing areas which was the same to our research (Figure 1 and Table 2).”  I am not sure I understand these two sentences. Clearer explanation is needed. The first is not really a correct sentence.

 Line 345.’ Vegetation C, N storage can depend on soil C pool and N availability’.  That is true, but the dependence is both ways. The soil C and N pools generally depend on litterfall from the vegetation.

 Overall, the results of the study seem reasonable and are a contribution to knowledge of primary producers in the Tibetan Plateau.

 Minor comments. The English is greatly improved in this version, but I make a number of suggestions below for improving the grammar and style.

 Line 11.  Maybe change ‘and where it is the important global terrestrial carbon (C) and nitrogen (N) pools, so’ to ‘and there are important global terrestrial carbon (C) and nitrogen (N) pools in that region, so’

Line 17. Change ‘above ground’ and ‘below ground’ to ‘above-ground’ and ‘below-ground’.

Line 23. Change ‘mean of foliar’ to ‘mean values of foliar’

Line 26. Change ‘it had’ to ‘there were’

Line 33. Change ‘productions’ to ‘productivity’

Line 37. Change ‘chinese’ to ‘Chinese’

Line 50. Change ‘were’ to ‘are’ and insert comma after ‘processes’

Line 59. Change ‘was close’ to ‘was a close’

Line 61. Change ‘which was increased’ to ‘which increased’

Line 66. Change ‘storge’ to ‘storage’

Line 92. Change ‘ration’ to ‘ratio’

Line 109. Change ‘Where’ to ‘where’ and ‘represented’ to ‘represent’

Line 116. Change ‘Itally’ to ‘Italy’

Line 122. I think this should be changed to ‘AGB was randomly collected and mixed together as a block, forming a composite sample’

Line 131. Change ‘Where’ to ‘where’

Line 142. Change ‘in atmosphere’ to ‘in the atmosphere’’

Line 162. Change ‘almost twice than that in AD’ to ‘almost twice that of AD’

Line 168. Change ‘empirically’ to ‘empirical’

Line 172. ‘modelling’ can be deleted.

Line 188. Change ‘was in’ to ‘were in’

Line 191. Change ‘was ranging’ to ‘ranged’

Line 203. Change ‘were shown’ to ‘are shown’

Table 1, Column 6. Change ‘specie’ to ‘species’

Line 229. Change ‘Empirical’ to ‘empirical’

Table 4.  Change ‘g C m2’ to ‘g N m2’ in columns 4, 5, and 7

Line 296. Change ‘Grassland biomass which comes from vegetations is’ to ‘Biomass from the vegetation component of grasslands is’

Line 298.  I think ‘formats’ should be ‘forms’

Line 301. Change ‘in fragile’ to ‘in the fragile’

Line 302. Change ‘had the similar’ to ‘had similar’

Line 303. Change ‘more’ to ‘greater’

Line 306. Change ‘always’ to ‘is always’

Line 316.  Change ‘BGB when’ to ‘BGB. When’. Also, I think ‘unity’ should be changed to either ‘linear’ or ‘in the same proportion’

Line 324. Change ‘caused environment unsteady was severe than’ to ‘caused environmental unsteadiness that was more severe than’

Line 331. Change ‘need’ to ‘needs’

Line 348. Change ‘was contributed’ to ‘contributed’

Line 351. Change ‘same to’ to ‘same as’

Line 354. Change ‘were important’ to ‘is important’

Line 363. Maybe change ‘which limited the biomass’ to ‘which limited the AD biomass’

Line 370. Change ‘When plant facing to’ to ‘When a plant faces’

Line 377. Change ‘Under stress environment’ to ‘Under a stressful environment’

Line 382. Change ‘is in low level’ to ‘is at a low level’

Line 396.  Change ‘while the highest value’ to ‘ , the highest value’

Reviewer 3 Report

Dear Authors,

the paper has been imporved a lot. I think it is ready for publication.

Author Response

Dear reviewer:

Thank you very much for your comments.